# Novel Treatment Strategies Utilizing Immune Reactions against Chronic Myelogenous Leukemia Stem Cells

**DOI:** 10.3390/cancers13215435

**Published:** 2021-10-29

**Authors:** Maiko Matsushita

**Affiliations:** Division of Clinical Physiology and Therapeutics, Faculty of Pharmacy, Keio University, Tokyo 105-8512, Japan; matsushita-mk@pha.keio.ac.jp

**Keywords:** chronic myelogenous leukemia, leukemic stem cells, immunotherapy

## Abstract

**Simple Summary:**

Although tyrosine kinase inhibitors (TKIs) are highly effective in the treatment of patients with chronic myelogenous leukemia (CML), leukemic stem cells (LSCs) are known to be resistant to TKIs. As a result, the application of immunotherapies against LSCs may cure CML.

**Abstract:**

Introduction of tyrosine kinase inhibitors (TKIs) has improved the prognosis of patients with chronic myelogenous leukemia (CML), and treatment-free remission (TFR) is now a treatment goal. However, about half of the patients experience molecular relapse after cessation of TKIs, suggesting that leukemic stem cells (LSCs) are resistant to TKIs. Eradication of the remaining LSCs using immunotherapies including interferon-alpha, vaccinations, CAR-T cells, and other drugs would be a key strategy to achieve TFR.

## 1. Introduction

Chronic myelogenous leukemia (CML) is a myeloproliferative neoplasm caused by t(9;22)(q34;q11) that produces the *BCR-ABL1* fusion gene, which encodes protein with dysregulated tyrosine kinase activity. CML progresses from the chronic phase (CP) to the accelerated phase (AP) and finally the blast crisis (BC) phase [1]. Targeted inhibition of BCR-ABL1 by tyrosine kinase inhibitors (TKIs), such as imatinib, dasatinib, nilotinib, bosutinib, and ponatinib can induce remission in patients with CP and prolong their survival by preventing transition to AP and BC [2,3,4,5]. However, single-cell transcriptomics have revealed that *BCR-ABL1*-positive leukemic stem cells (LSCs) persist even in patients with major molecular responses that can proliferate and differentiate into CML cells [6,7]. CML-LSCs are resistant to TKIs because they are quiescent, and their survival is not dependent on ABL1 kinase activity [8,9,10]. Therefore, the addition of other treatment methods against CML-LSCs might be needed to achieve a cure in CML patients [11].

On the other hand, after cessation of TKI, about half of the patients could maintain a major molecular response (MMR) without TKIs, which is called treatment-free remission (TFR) [12,13,14]. In these patients, immune cells could have suppressed or eradicated persisting LSCs. We previously identified CXorf48 as an immunological target that is highly expressed in CML-LSCs. Antigen-specific cytotoxic lymphocytes (CTLs) were detected in the peripheral blood of patients who remained in molecular complete remission (CR) after discontinuation of imatinib, suggesting that CTLs could suppress the expansion of CML-LSC in these patients [15].

Taken together, the combination of TKIs and immunotherapy would be a powerful treatment strategy for achieving a cure in CML patients (Figure 1). Recently, the results of TKI cessation trials have provided useful information on the interactions of the immune system with CML-LSCs [16,17,18,19,20].

Here, we review the immunological status of CML patients, the immunomodulatory effects of TKIs, and current attempts to enhance immune reactions against LSCs in CML patients.

## 2. Immune Status of CML Patients

### 2.1. Immunogenecity of CML-LSCs

CML-LSCs are enriched in CD34^+^CD38 low cells [21,22]. Like other cancer stem cells of solid tumors, LSCs can escape from host immunity by several mechanisms [23,24]. Tarafdar et al. found that *IL-4* gene expression was upregulated in CML-LSCs, which suppresses class II transactivator (CIITA) transcription and subsequently expression of major histocompatibility complex-II (MHC-II) in CML-LSCs compared to normal hematopoietic stem cells (HSCs) [25]. The downregulation of MHC-II might reduce antigen presentation against immune cells. Levescot et al. reported that IL-33 binds to the IL-1 receptor ST2 on CML-LSCs and contributes in the maintenance of LSC and imatinib resistance [26]. Therefore, CML-LSCs conserve relatively low immunogenicity, and a combination of several therapeutic methods would be required for depleting these cells by immunotherapies.

### 2.2. Immune status in CML Patients at Diagnosis

During the diagnosis of CML, patients’ immune cells are derived from CML-LSC, and aberrations of both innate and acquired immunity are observed in these patients [27,28,29]. Hughes et al. investigated the peripheral blood of 21 patients with CML-CP from diagnosis until treatment with TKIs. A decrease in the number of natural killer (NK) cells and cytotoxic T-lymphocytes (CTLs), and impaired responses to leukemia-associated antigens WT1, proteinase 3, BMI-1, and preferentially expressed antigen of melanoma (PRAME) were observed at diagnosis [30]. Moreover, the tumor microenvironment of CML cells is immune-inhibitory, characterized by an increase in the number of immunosuppressive cells such as myeloid-derived suppressor cells (MDSCs) and regulatory T cells (Tregs) or upregulation of programmed death-1 (PD-1) on CD4/CD8 T cells (Figure 2) [30,31,32]. Immune dysfunction was stronger in high-risk patients [33]. In total, anti-CML immune responses were suppressed in patients with CML at diagnosis.

### 2.3. Immunomodulatory Effects of TKIs in CML

ABL-TKIs not only directly kill CML cells, but also exert immune-modulatory effects. In vitro studies have used peripheral blood mononuclear cells (PBMCs) from healthy donors, showing that TKIs suppress the proliferation of T cells, B cells, and NK cells at therapeutic doses [34,35,36]. However, other clinical studies have shown that TKIs could restore immune inactivated status in CML patients and indirectly suppress CML cells. Since direct killing is less frequently observed in CML-LSCs, these immunomodulatory effects are important for the eradication of CML-LSCs.

#### 2.3.1. Imatinib

Tanaka et al. have reported that imatinib induces apoptosis of Tregs by its off-target effects on LCK at its therapeutic doses, because imatinib targets a variety of tyrosine kinases other than ABL1, including KIT, PDGFR, LCK, NQO2, and DDR1 [37]. They also reported that peripheral blood of imatinib-treated CML patients in complete molecular remission (CMR) had selective depletion of FoxP3^hi^CD45RA^−^ effector T reg (eTreg) cells, which are in a highly proliferative state and dependent on the TCR signal. They speculated that LCK inhibition attenuates the TCR signal, thereby inducing signal-deprived cell death. This selective depletion of eTregs was not observed in CML patients without CMR. They then stimulated PBMCs of healthy donors with peptides from Melan-A/MART-1, a melanoma-specific tumor antigen, with or without a therapeutic dose of imatinib, and showed that imatinib treatment expanded Melan-A-specific CD8^+^ T cells accompanied by the depletion of eTregs. An increase in the effector and memory CD8^+^ T cells and reduction of exhausted CD8^+^ T cells with PD-1 and LAG-3 expression was observed in CMR patients. Kreutzman et al. showed that the proportion and function of CD56^+^CD16^+^NK cells increased withimatinib therapy in CML patients [38]. Taken together, the impaired anti-CML immunity can be recovered by imatinib.

#### 2.3.2. Dasatinib

Dasatinib is known to induce the expansion of large granular lymphocytes (LGLs) [39]. Kreutzman et al. reported the development of dasatinib-associated lymphocytosis at an average of three months after the start of treatment and its persistence throughout the therapy. The expanded lymphocytes contained both CD3^+^CD8^+^ effector memory T-cells and CD3^−^CD16^+^CD56^+^ NK cells [40]. Importantly, the median progression-free survival or overall survival was superior in patients with increased LGL expansion compared to those without [41]. Dasatinib is also known to affect Tregs and NK cells. In a prospective phase II clinical trial, D-first study, peripheral blood of 52 patients with newly diagnosed CM-CP was observed for a minimum of 36 months after starting daily administration of 100 mg dasatinib. Treg decreased with dasatinib therapy, and the proportion of Tregs at 12 months of dasatinib treatment was associated with the achievement of deep molecular response (DMR). Treg inhibition was also correlated with an increase in CTL counts, NK cell count, and NK cell differentiation [42]. The effects on subsets of NK cells have also been analyzed in several studies. Ishiyama et al. recently reported that CD56^−^ NK cells with PD-1 expression were induced by chronic activation through CMV reactivation in CML or Ph + ALL patients during dasatinib therapy [43]. Interestingly, the expansion of CD56^−^ NK cells was associated with a higher rate of DMR, and Wei et al. also reported that dasatinib increased the percentage of Th1 and CD8 T cells and decreased Treg cell levels in the peripheral blood of CML patients who responded to therapy [44]. Overall, dasatinib seems to have the potential to enhance both innate and adaptive immunity to eradicate CML-LSCs in some patients. Clinical trials (ClinicalTrials.gov Identifier: NCT04991532) are also ongoing in China to assess the effects of dasatinib on NK cells and T cells of CML-CP patients to compare the immunomodulatory effects of TKIs in clinical settings.

#### 2.3.3. Nilotinib

In vitro experiments have shown that nilotinib inhibited the proliferation of Tregs; however, it did not induce apoptosis of Tregs at a clinical dose, which is different from imatinib [45]. Nilotinib suppresses the differentiation of dendritic cells from peripheral blood monocytes of healthy donors with reduced surface expression of CD83 and CD14. IL-12p70 production from DCs was also impaired after treatment with nilotinib, suggesting that nilotinib has immunosuppressive effects [46].

#### 2.3.4. Bosutinib

Although little is known about the immunomodulatory effects of bosutinib, there is a report analyzing the immune phenotype of newly diagnosed CML patients treated with imatinib (*n* = 20) or bosutinib (*n* = 13) [38]. No major immunological changes were observed during bosutinib treatment compared to imatinib treatment, possibly because of its narrow spectrum of target kinases.

#### 2.3.5. Ponatinib

Recently, Leonard et al. reported a combination therapy of blinatumomab and ponatinib for Ph^+^ ALL. They showed that ponatinib suppressed the expansion of T cells and the production of IFN-γ induced by blinatumomab. They found that this effect was mediated by inhibiting the phosphorylation of LCK. These results suggest that ponatinib might suppress anti-leukemia immunity in CML patients [47].

Table 1 summarizes the immunomodulatory effects of TKIs. However, there is little information on the immunomodulatory effects of novel TKIs, including asciminib, which binds to the myristate binding pocket on the kinase domains of ABL1 and BCR-ABL1, and which are effective in patients who are resistant to other TKIs [48], and it would be of interest to assess the impact of these drugs on immune cells in the future.

## 3. TFR and Immune Cells

### 3.1. TKI Discontinuation Trials

Introduction of TKIs has improved the overall survival of patients with CML-CP. Since TKIs are not able to eradicate CML-LSCs, life-long administration is required to maintain a DMR. However, long-term adverse effects, including cardiovascular disease, decreased quality of life, and economic burdens can be problematic in some patients [49,50,51]. Therefore, clinical trials of discontinuing TKIs have been conducted in patients with sustained DMR [52,53,54]. The first discontinuing trial, a STIM study, enrolled patients on imatinib therapy for a minimum of three years in whom BCR-ABL1 transcripts were undetectable for at least two years, showing that 40% of patients stayed in DMR after cessation of imatinib [55]. After this trial, various kinds of discontinuation trials have been reported in CML-CP patients using imatinib or second generation TKIs [56,57,58,59,60,61]. Indeed, about half of the patients who stopped taking TKIs remained in the MMR status.

### 3.2. Role of Immune Reactions against CML-LSC in TFR

Genomic DNA-based PCR, in TKI discontinuation clinical trials, revealed that patients continued to harbor the *BCR-ABL1* gene after discontinuation of imatinib, even in patients with TFR [62,63], suggesting that the patients’ immune systems suppressed the residual CML-LSC and prevented disease recurrence. Mainly NK cells and CD8^+^ cells activated by dendritic cells target CML-LSCs.

The analysis of some clinical trials discontinuing TKIs, such as TWISTER or EURO-SKI, showed that the prior use of interferon-alpha (IFN-α) was correlated with a higher probability of achieving TFR [58,59]. This might be because of immune reactions activated by IFN-α controlled residual CML-LSCs. Burchert et al. have reported that IFN-α maintenance therapy after imatinib withdrawal, mediated expansion of anti-PRI CTL in four out of seven assessable patients; this was associated with stable disease, suggesting that enhancement of effector T cells by IFN-α contributed to disease control [64]. In the TKI-DISC trial, sustained TFR was associated with increased innate CD8^+^ cells [65]. In CML8 and CML10 trials, an increased number of NK cells and a decreased number of immunosuppressive cells, including Tregs and MDSCs, were associated with TFR [66]. The phenotype of NK cells was also analyzed in these trials. For example, haplotypes of killer-cell immunoglobulin-like receptors (KIRs) are correlated with TFR [67]. Patients who are homozygous for the KIR A haplotype have better molecular responses and higher rates of successful TFR compared with other KIR haplotypes [68]. It has also been reported that the KIR2DL5B genotype correlates with a lower probability of MMR and DMR in patients receiving TKI therapy, as well as a lower chance of achieving TFR [69]. These data suggest that the anti-CML activity of NK cells is defined by the KIR haplotype, and also the polymorphisms of HLA molecules to which they correspond. Based on these observations, immunological parameters that distinguish TFR-achievable patients have been investigated; however, the precise markers have not been determined.

## 4. Immunotherapies against CML-LSCs

Since allogeneic stem cell transplantation or donor lymphocyte infusion could cure CML patients, CML-LSCs seem to be sensitive to attack by immune cells [70,71]. Although immunological surveillance is impaired in CML patients, as mentioned above, recent progress in the development of novel immunotherapies against cancers is also applicable to CML. For this purpose, several kinds of immunotherapies, including IFN-α, are under investigation (Table 2).

### 4.1. IFN α

IFN-α had been used as a standard therapy in transplant-ineligible CML-CP patients before introduction of TKIs [72]. Several modes of action, including induction of apoptosis and suppression of cell proliferation, are known [73,74]. In addition, IFN-α activates immunity by enhancing the proliferation of NK cells, CD8^+^T cells, and mature dendritic cells [75]. Paquette et al. showed that PBMCs obtained from CML patients could be differentiated into dendritic cells by adding IFN-α and GM-CSF with subsequent proliferation of T cells [76]. They also showed that bone marrow from IFN-α-responded patients contained a higher number of dendritic cells. Furthermore, type I IFNs (IFN-α,β) increase the expression of both HLA class I and antigen expression and stimulate CTLs [77].

These multiple effects of IFN-α in eradicating CML stem cells have recently attracted attention [78,79]. Since IFN-α synergizes with TKIs, a combination of IFN-α and TKIs are now examined in several clinical trials. The French STI571 prospective randomized trial (SPIRIT) compared patients treated with imatinib 400 mg versus imatinib 600 mg, imatinib 400 mg plus cytarabine, and imatinib 400 mg plus pegylated interferon alpha2a (PegIFN-α2a) for the front-line treatment of CML-CP patients [80]. The rates of MMR at 12 months and DMR (MR^4^) over time were significantly higher with the combination of imatinib 400 mg and PegIFN-α2a than with imatinib 400 mg alone. Additionally, long-term follow-up results were reported and overall survival after a median follow-up of 15 years did not differ among groups [81]. Although discontinuation of imatinib after sustained DMR was not planned in this trial, 350 patients discontinued their treatment. The proportion of patients with TFR was similar among all the treatment groups. However, IFN-α was discontinued in 45% of the patients, and the median duration of Peg IFN-α2a therapy was 1.1 years. This might be too short to induce immune reactivity, which could prevent relapse after treatment cessation. A longer period of IFN treatment should be considered in future trials. German study IV [82], one of the largest academic trials, included 1551 newly diagnosed CP-CML patients, and showed that the imatinib (800 mg → 600 mg) group was superior to the imatinib plus IFN group in MMR rates at 12 months (55% vs. 33%). They did not use peg IFN, and most of the patients discontinued IFN therapy due to side effects. The use of peg IFNα would be necessary to obtain long-term benefits from combination therapy.

Recently, clinical trials using a combination of 2nd generation TKIs and IFN have also shown that combination therapy of second generation TKIs and peg IFN resulted in a higher rate of DMR [83]. In NordCML007 [84], a single arm phase II clinical trial, CML-CP patients were first treated with dasatinib for three months, and then pegylated IFN-α2b (peg IFN-α2b:15 µg→25 µg/week) was added for 12 months. The results showed a higher rate of DMR in the combination group at 12 or 24 months than that reported in the DASISION trial. Recently, long-term follow-up data were presented [85], showing that the rate of MMR and DMR remained high even after 5 years without severe adverse effects (rates of MR^3.0^, MR^4.0^ and MR^4.5^ were 84.6%, 64.1% and 51.3% respectively). In the NiloPeg trial, 41 patients were previously treated with peg IFN-α2a (90 µg) for 1 month, followed by combination therapy with peg IFN-α2a (45 µg) and nilotinib (600 mg daily) for 24 months. Seven patients achieved MR^4.5^ at 12 months, although hematological and hepatic adverse effects (>grade3) occurred in 24% of patients [86].

These results suggest that this combination provides higher DMR in the early phase, and dose adjustment to reduce the toxicity of peg IFN with maximum immunological effects might be needed to obtain an extension in overall survival.

### 4.2. Vaccination

CML cells express both leukemia-specific antigen derived from the BCR-ABL fusion protein and leukemia-associated antigens, including Wilms tumor 1(WT1), proteinase 3 (PR1), and preferentially expressed antigen of melanoma (PRAME) [87,88,89,90]. Molldrem et al. have reported a strong correlation between the presence of PR1-specific T cells and clinical responses after IFN-α and allogeneic BMT in CML patients, suggesting that CTLs against CML antigen play some role in eradicating CML-LSCs [91]. Therefore, enhancement of T cell immunity by vaccination with CML-expressing antigens has been considered as one of the immunotherapies against CML. Since high tumor burden is difficult to eradicate only by vaccination, CML patients receiving TKI therapy with minimal residual disease might be good candidates for vaccination.

#### 4.2.1. BCR-ABL1peptides

The epitopes containing the breakpoint of the P210 BCR-ABL1 fusion protein are attractive leukemia-specific targets, since normal cells do not express this oncogenic protein [92,93]. In a phase II clinical trial, ten patients with CP who had been on imatinib therapy (median 62 months) were treated with imatinib and vaccinated with a mixture of peptides derived from both b3a2 and b2a2 every two weeks for four weeks (Weeks 0, 2, 4, and 6), then every three weeks for one week (Week 9), and then monthly for 10 months [94]. One log reduction in BCR-ABL1 transcript levels was observed in three patients without severe adverse effects. In another clinical trial [95], 16 CML-CP patients who underwent stable disease with imatinib or IFN-α therapy were injected with six weekly multiple p210-b3a2-derived peptide vaccines and adjuvants (CMLVAX100). All but one patient with vaccine-induced positive CD4 immune response presented a reduction in *BCR-ABL1* transcripts. These results suggest that BCR-ABL1-derived peptide vaccine may improve the molecular response.

#### 4.2.2. Leukemia-Associated Antigens

CML-associated antigens include WT1, PR1, and PRAME. These antigens are expressed in some normal cells; however, their expression levels are higher in CML cells and they could be therapeutic targets recognized by CTLs [90,96]. Oji et al. reported that CML patients who had persistent BCR-ABL transcripts after imatinib were treated with HLA-A*24:02-restricted modified WT1 peptide vaccine every two weeks with imatinib, and a reduction in BCR-ABL mRNA was observed to the detection limit [97]. The WT1-specific tetramer assay revealed that the decrease in BCR-ABL transcripts was associated with an increase in the frequency of WT1-specific CTLs in the patient’s peripheral blood. Qazilbash et al. conducted a phase I/II clinical trial using HLA-A*02:01 restricted PR1 peptide vaccination without other treatments [98]. Sixty-six patients with active myeloid malignancies, including 13 CML patients, were subcutaneously vaccinated every three weeks. Treatment response was observed in 12 (22%) of 53 patients vaccinated with active disease, including two CML patients who achieved molecular responses. An increase in the percentage of PR1-specific tetramer-positive CTLs in peripheral blood was correlated with clinical response, although that of cytomegalovirus-specific CTL was not. They also found that PR1-CTL was significantly enriched in central memory cells, which suggested that the PR1 peptide vaccine elicited long-lasting immune responses. Although this vaccination study did not combine TKI therapy, it showed promising results, suggesting that vaccination after TKI discontinuation could be a treatment option to cure residual CML-LSC.

#### 4.2.3. Cellular Vaccines

In addition to peptide vaccinations, cellular vaccines using BCR-ABL+ cells could evoke anti-CML immunity by multiple epitopes derived from CML cells [99]. One method of vaccination includes the use of dendritic cells induced from BCR-ABL+ monocytes of patients. In a phase I/II study, autologous DC vaccines were subcutaneously injected into CML-CP patients who did not achieve an adequate cytogenetic response after treatment with IFN-α or imatinib. Improvement of the cytogenetic/molecular response was detected in four of ten patients. In three of these patients, T cells recognizing leukemia-associated antigens were detected. In another method, the CML cell line, K562, which was genetically modified to express GM-CSF, was administered [100]. The K562 vaccine could elicit T cell response against CML cells in two out of three patients with stable remission after DLI.

### 4.3. Immune Checkpoint Inhibitors

The emergence of immune checkpoint inhibitors, including anti-cytotoxic T lymphocyte antigen 4 (CTLA4) antibody, anti-programmed death 1 (PD1) antibody, and anti-PD ligand 1(PDL1) antibody have changed cancer treatments [101,102,103], although their use is limited in hematological malignancies despite Hodgkin’s lymphoma [104]. Mumprecht et al. reported that CML-specific T cells get exhausted and express high levels of PD-1 in a CML mouse model [105]. PD-L1 was also expressed on the surface of granulocytes from mice with CML. They showed that blocking the PD-1 signal using PD-1–deficient recipient mice or via the administration of the PD-L1 antibody restored CML-specific T-cell tolerance and prolonged survival of CML-BC mice. In CML patients, upregulation of PD-1 on both CD4+ T cells and CD8^+^ T cells at diagnosis suggests that T lymphocytes were exhausted after chronic stimulation by CML antigens. After administering TKI, the expression level of PD1 decreased with concomitant recovery of leukemia-associated antigen-specific CTL activities, such as IFN-γ production [30]. These findings indicate that a blockade of the PD1/PD-L1 pathway might restore anti-CML immunity. Moreover, recent analysis using next-generation sequencing has revealed that BCR-ABL1-dependent or-independent mutations exist in patients with poor response to TKIs [106,107], suggesting that neoantigens other than known CML antigens could be targeted by immune checkpoint inhibitor (ICI)-activated CTLs which reside in these CML patients. Therefore, several clinical trials using ICIs in patients with CML are ongoing. Phase II clinical trials of combination therapy of pembrolizumab and TKIs (dasatinib, imatinib, or nilotinib) in CML-CP patients with persistent minimal residual disease is ongoing (Clinical Trials.gov Identifier number: NCT03516279).

### 4.4. Chimeric Antigen Receptor-T Therapy

Chimeric antigen receptor (CAR)-T cells can bind to surface antigens of cancer and strongly eradicate target cells. Immunotherapy using CAR-T cells have rapidly been introduced to cancer treatment and has shown unprecedented clinical effects [108]. In particular, CAR targeting CD19 is genetically introduced into patients’ T lymphocytes and is used for the treatment of B cell malignancies [109,110,111]. Anti- B cell maturation antigen (BCMA) CAR-T therapy is also effective in patients with multiple myeloma and has been approved by the FDA [112,113]. With regard to CML, IL1 receptor-associated protein (IL1RAP) is one of the candidate cell surface antigens for CAR-T cells [114]. IL1RAP is a receptor for IL-1 and IL-13 and is expressed by CML-LSCs, but not by normal HSCs. Warda et al. combined scFv of monoclonal antibody against IL1RAP, and CD3z domain followed by CD28 and 4-1BB into a lentiviral construct carrying the iCASP9 safety cassette [115]. These CAR-T gene-transduced T cells recognize CML cells, and only a portion of monocytes in normal blood cells as off-target effects. In a tumor xenograft murine model, IL1RAP CAR T cells eliminated K562 and KU812 cells. They could also generate IL1RAP CAR-T cells using lymphocytes from TKI-resistant CML patients. Autologous IL1RAP CAR T cells exhibit specific lysis of CML cells. They are currently conducting clinical trials of administering CAR gene-transduced patients’ lymphocytes to CML patients (Clinical Trials.gov Identifier number: NCT02842320.). Several groups have identified other surface markers, such as CD26, CD25, CD33, CD44, and CD123 [116,117]. These antigens are highly expressed in CML-LSCs compared to normal HSCs, and they could also be targets of CAR-T cells for CML patients. Recently, Zhou et al. reported the successful generation of CD26 CAR-T cells targeting CML LSCs [118]. CD26 CAR-T cells showed cytotoxicity against CD26-expressing K562 cells and CD34^+^CD26^+^ cells from CML patients in vitro and in vivo. Off-target cytotoxicity against activated lymphocyte was observed, which was within the normal reference range. TKI-non-responders, TKI-intolerant CML patients, or AP/BC CML patients would be eligible for this therapy.

### 4.5. Other Immunomodulations

As mentioned in Section 2.1, MHC-II molecules are downregulated in CML-LSCs compared to normal HSCs, which might cause immune escape [25]. Upregulated *IL-4* gene expression in CML-LSCs could suppress CIITA transcription and MHC-II expression. Targeting of the downstream JAK-STAT pathway by ruxolitinib [119], a JAK1/2 inhibitor, recovered MHC-II expression and increased the proliferation of CD4^+^ cells against CML-LSC. Combining ruxolitinib with other immunotherapies to enhance immunogenicity might be useful to potentiate the immunological eradication of CML-LSCs. Zhang et al. have also shown that IL-1 receptors, IL-RAP and IL-1R1, are upregulated in CML-LSCs [120]. Inhibition of IL-1 signaling by IL-1 receptor antagonist (IL-1RA) inhibited the expansion of CML-LSCs in combination with TKIs.

Recently, combination therapies of TKIs and other agents, such as venetclax, BCL-2 inhibitor, or piogliteazone, peroxisome proliferators-activated receptor-gamma (PPARg) agonist, are in clinical trials [121]. These drugs may not only render CML-LSCs sensitive to TKIs, but also have the potential to activate immune cells. Venetclax directly blocks the anti-apoptotic BCL-2, which is overexpressed in CML-LSCs and induces cellular apoptosis [122]. Moreover, Lee et al. recently reported that venetclax enhances T cell effector function against acute myeloid leukemia cells through production of reactive oxygen species [123]. Another example is pioglitazone, which is known to sensitize CML-LSCs to TKIs by repressing the transcription of *STAT5*, which up-regulates stemness-related genes [124]. Fu, et al. have shown that pioglitazone could also induce IFN-γ production in tumor-infiltrating invariant natural killer T from both human patients and mouse melanoma models through alteration of metabolic status in these cells [125], suggesting that the immune-activating effects of these drugs might be related to the efficacy of the combination therapies in CML patients.

## 5. Conclusions

Remaining CML-LSCs after TKI treatment are considered to be immunogenic, however, the immune system in CML patients is impaired at diagnosis, and restored by TKI treatment to some extent. After discontinuation of TKIs, NK cells or CTLs could control the expansion of residual CML-LSCs in some patients. It is feasible to enhance immunity against CML-LSCs by using immunotherapeutic methods combined with TKI and increase the probability of TFR (Figure 3). Since immune status varies depending on the disease status, type of TKIs, and characteristics of the host immune system, the patient-oriented combination strategy should be clarified for the achievement of TFR in the future. In addition, predictive markers of immune function are needed to distinguish patients who are eligible for TKI discontinuation in the CML-CP state. Clinical trials with the detailed analysis of the immune status of patients are important.

PegIFN-α, pegylated interferon alfa; CAR-T cells, chimeric antigen receptor-T cells; IL-1RA, Interleukin-1 receptor antagonist; CCR, complete cytogenetic remission; MMR, major molecular remission; DMR. deep molecular remission; TFR, treatment-free remission.

## Figures and Tables

**Figure 1 cancers-13-05435-f001:**
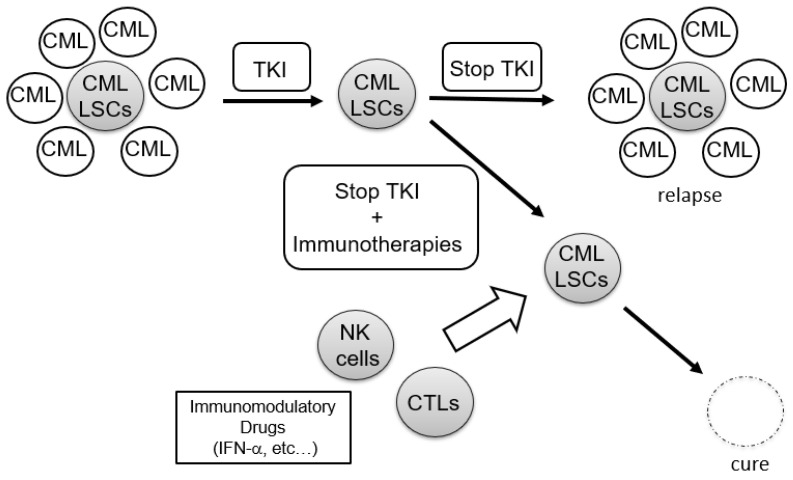
Eradication of CML-LSCs by immunotherapies after TKI therapy. CML, chronic myeloid leukemia; LSCs, leukemic stem cells; TKI, tylosine kinase inhibitor; NK cells, natural killer cells; CTLs, cytotoxic T cells; IFN-α, interferon alpha.

**Figure 2 cancers-13-05435-f002:**
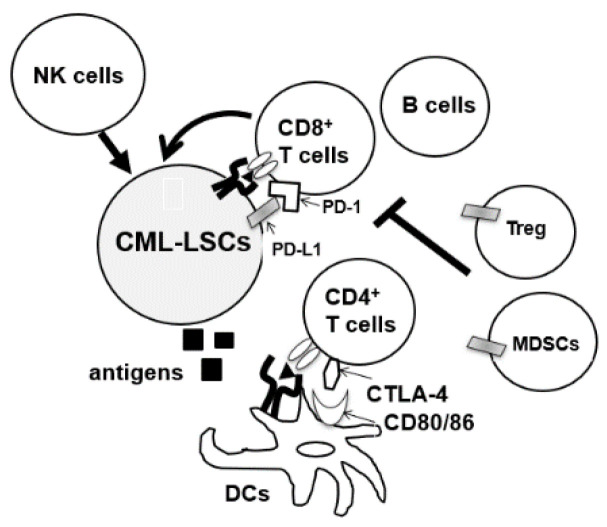
CML-LSCs and immune cells in the tumor microenvironment. NK cells, natural killer cells, MDSCs, myeloid-derived suppressor cells; DCs, dendritic cells, CTLA4, cytotoxic T-lymphocyte-associated protein 4.

**Figure 3 cancers-13-05435-f003:**
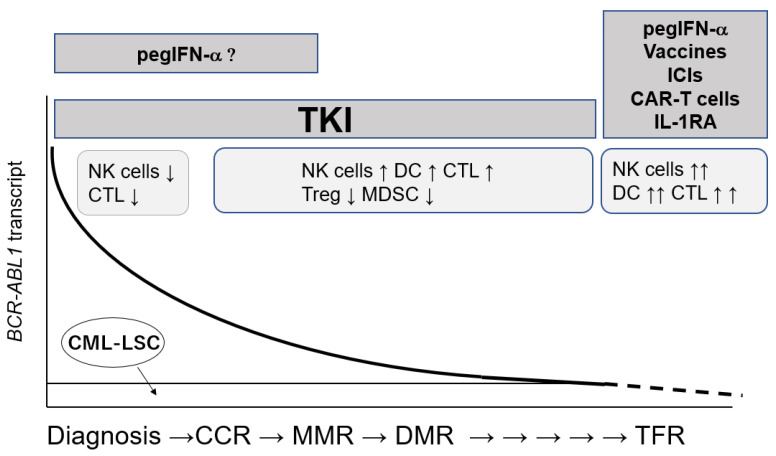
Clinical course of CML-CP patients and immunotherapies combined with TKI therapy to achieve TFR.

**Table 1 cancers-13-05435-t001:** Immune-modulatory effects of TKIs.

TKIs	Immune-Modulatory Effects
Imatinib	Treg ↓, effector and memoriy CD8^+^T cells ↑, exausted T cells ↓, NK cells ↑
Dasatinib	LGL (CD8^+^T cells or NK cells) ↑,Treg ↓
Nilotinib	Treg ↓DC (differentiation, IL-12 production) ↓
Bosutinib	No change in T cells, monocyte, and granulocytes
Ponatinib	CD8^+^T cells (proliferation IFN-g production) ↓

T reg, regulatory T cells; NK cells, natural killer cells; LGL, large granular lymphocytes; DC, dendritic cells. ↑, increase ↓, decrease.

**Table 2 cancers-13-05435-t002:** New immunotherapies for targeting CML-LSC.

Immunotherapies	Targets	Mode of Actions
IFN-α	CTL, NK cells CML-LSC	Activation of immune cells Upregulation of HLA class I
Vaccines	CML-specific antigens (BCR-ABL) CML-associated antigens (PR1, WT1, PRAME) CML cells (K562-GM-CSF)	Activation of CML-specific CTLs
Immune checkpoint inhibitors	PD-1 on CTLs and NK cells PD-L1 on CML-LSCs	Reactivation of exhausted immune cells
Chimeric antigen receptor-T cells	CML-LSC surface antigens IL1RAP, CD26, CD25, CD33, CD44, CD123	Target-dependent lysis of CML-LSCs

HLA, human leucocyte antigen; PR1, a peptide derived from protenase 3; WT1, Wilms tumor 1; PRAME, preferentially expressed antigen of melanoma; GM-CSF, Granulocyte Macrophage colony-stimulating Factor PD-1, programmed cell death protein 1; PD-L1, Programmed cell Death 1-Ligand 1.

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
