# Peer review of "Novel Treatment Strategies Utilizing Immune Reactions against Chronic Myelogenous Leukemia Stem Cells"

_cancers, 2021, doi:10.3390/cancers13215435_

Round 1
Reviewer 1 Report
It would be interesting if the authors describe the mechanism of sustained complete remission in patients treated just with TKI without any immunotherapy. Do the CML-LSC in these patients more responsive to innate immunity?
Author Response
I appreciate the time and effort each of the reviewers has dedicated to providing insightful feedback on ways to strengthen my review paper. Thus, it is with great pleasure that I resubmit my article for further consideration. I have incorporated changes that reflect the detailed suggestions you have graciously provided. I also hope that my edits and the responses I provide below satisfactorily address all the issues and concerns you and the reviewers have noted.
I agree with the reviewers’s suggestion. In fact, analysis of immune cells in patients during treatments with only dasatinib suggests that expanded NK cells by dasatinib could eradicate LSCs and induce DMR in some patients (described in Section 2.3.2 line 128-129). Although there is no direct evidence that these NK cells could react to CML-LSCs, I have clarified the point by adding the explanation below (line 131-133).
‘Overall, dasatinib seems to have the potential to enhance both innate and adaptive immunity to eradicate CML-LSCs in some patients.’
Reviewer 2 Report
In this interesting paper Dr Matsushita discusses several strategies to overcome BCR-ABL leukemic stem cells positivity in patients with chronic myelogenous leukemia (CML) after stopping therapy with tyrosine kinase inhibitors (TKIs), with the goal to improve cure rate in this setting. Recently, Mu H et al have published a similar review (reference 11)
Some major considerations to include:
1. The immunomodulatory effects of ponatinib in CML
2. If the possible synergism of TKIs and venetoclax combination in order to a better control of CML could be in part explained because of effects against quiescent leukemic stem cells. Similarly, the possible role of PPAR-gamma agonists and ongoing clinical trials should be discussed.
3. Finally, it is worth mentioning if there are data from new approaches to the treatment of CML such as asciminib in this setting.
Minor comments:
- In lines 25, 140 and 141 bosutinib should be corrected.
- To define abbreviations in Figures 1 and 2, and Table 1.
Author Response
I appreciate the time and effort each of the reviewers has dedicated to providing insightful feedback on ways to strengthen my review paper. Thus, it is with great pleasure that I resubmit my article for further consideration. I have incorporated changes that reflect the detailed suggestions you have graciously provided. I also hope that my edits and the responses I provide below satisfactorily address all the issues and concerns you and the reviewers have noted.
- The immunomodulatory effects of ponatinib in CML
I agree with the reviewer’s comments. Although there is no report on immunomodulatory effects of ponatinib in CML patients, explanation was added as section 2.3.5. mentioning a recent paper of using ponatinib in Ph+ ALL (line 154-158) as follows.
2.3.5 Ponatinib
Recently, Leonard et al. reported a combination therapy of blinatumomab and ponatinib for Ph+ ALL. They showed that ponatinib suppressed the expansion of T cells and production of IFN-g, induced by blinatumomab. They analyzed that this effect was mediated by inhibiting the phosphorylation of LCK. These results suggest that ponatinib might suppress anti-leukemia immunity in CML patients [47].
- If the possible synergism of TKIs and venetoclax combination in order to a better control of CML could be in part explained because of effects against quiescent leukemic stem cells. Similarly, the possible role of PPAR-gamma agonists and ongoing clinical trials should be discussed.
I did not include venetoclax and PPAR-gamma agonist since this review mainly focuses on immunological aspects of treatments. However, I agree to mention these drugs, since it seems that there are possibilities that these drugs also have immunological effects on LSCs. I added some explanation in Section 4.5 (line 402-415) .
Recently combination therapies of TKIs and other agents such as venetclax, BCL-2 inhibitor, or piogliteazone, peroxisome proliferators-activated receptor-gamma (PPARg) agonist, are in clinical trials [121], and these drugs may not only render CML-LSCs sensitive to TKIs, but also have the potential to activate immune cells.Venetclax directly blocks the anti-apoptotic BCL-2, which is overexpressed in CML-LSCs and induces cellular apoptosis [122]. Moreover, Lee et al. recently reported that venetclax enhances T cell effector function against acute myeloid leukemia cells through production of reactive oxygen species [123]. Pioglitazone, which is known to sensitize CML-LSCs to TKIs by repressing the transcription of STAT5, which up-regulates stemness-related genes [124]. Fu, et al. have shown that pioglitazone could also induce IFN-γ production in tumor-infiltrating invariant natural killer T from both human patients and mouse melanoma models through alteration of metabolic status in these cells [125], suggesting that the immune-activating effects of these drugs might be related to the efficacy of the combination therapies in CML patients.
- Finally, it is worth mentioning if there are data from new approaches to the treatment of CML such as asciminib in this setting.
As the reviewer pointed out, it would be necessary to discuss about novel TKIs in this review. Although there is almost no information on immunological impacts of these asciminib, I added some sentences in Section 2.3 (line 160-164).
Table 1 summarizes the immunomodulatory effects of TKIs. However, there is little information on the immunomodulatory effects of novel TKIs, including asciminib, which binds to the myristate binding pocket on the kinase domains of ABL1 and BCR-ABL1, which are effective in patients who are resistant to other TKIs [48], and it would be of interest to assess the impact of these drugs on immune cells in the future.
Minor comments:
- In lines 25, 140 and 141 bosutinib should be corrected.
- To define abbreviations in Figures 1 and 2, and Table 1.
I appreciate the reviewer to point out several points above.
Those points were corrected according to reviewer’s comments.

Reviewer 3 Report
Comments for the authors:
In the paper, the authors outline the relationship between the treatment for chronic myeloid leukemia (CML) using tyrosine kinase inhibitors (TKIs) and the influence on immunotherapies against leukemic stem cells (LSCs). Overall, although the manuscript is informative, to provide a better understanding of the topic for a broad readership, I suggest that the authors provide a table or a simplified figure that summarizes the study’s findings. In addition, several minor issues have been noted, which should be addressed by the authors before the manuscript is accepted for publication. Some suggestions are provided below:
- Given that this is a review article on the treatment of CML LSCs using TKIs and the influence on immune response regulation, the title of the manuscript should be revised to encompass the scope of the information presented.
- The first mention of molecular CR in line 38 is an abbreviation, which needs to be corrected.
- In addition to causing resistance to tyrosine kinase inhibitors (TKIs), are there other characteristics of leukemic stem cells that can help define this subset of cells (such as a molecular or surface marker)?
- Provide a simple illustration describing the immune status and the associated factors and tumor microenvironment in patients with CML.
- In paragraph 2.3 describing the TKI for Bcr-Abl, please provide a table with relevant information regarding its immunomodulatory potential and effects.
- I would like to know whether the combination of TKIs and immunotherapy is a good treatment strategy; if a patient harbor mutant Bcr-Abl, is combinatorial immunotherapy a potentially useful approach? This may also be a topic worthy of discussion, which can be discussed in the revised manuscript.
Author Response
I appreciate the time and effort each of the reviewers has dedicated to providing insightful feedback on ways to strengthen my review paper. Thus, it is with great pleasure that I resubmit my article for further consideration. I have incorporated changes that reflect the detailed suggestions you have graciously provided. I also hope that my edits and the responses I provide below satisfactorily address all the issues and concerns you and the reviewers have noted.
- Given that this is a review article on the treatment of CML LSCs using TKIs and the influence on immune response regulation, the title of the manuscript should be revised to encompass the scope of the information presented.
(new)
According to variable suggestion of the reviewer, title was changed as follows.
Novel treatment strategies utilizing immune reactions against chronic myelogenous leukemia stem cells.
(old)
Immunotherapy against chronic myelogenous leukemia stem cells
- The first mention of molecular CR in line 38 is an abbreviation, which needs to be corrected.
According to reviewer’s suggestion, abbreviation was explained in line 38.
complete remission (CR)
- In addition to causing resistance to tyrosine kinase inhibitors (TKIs), are there other characteristics of leukemic stem cells that can help define this subset of cells (such as a molecular or surface marker)?
As mentioned in line 56, CML-LSCs are contained in CD34+CD38low fraction. Other surface markers, such as CD26, CD25, CD33, CD44, and CD123 can be found in the CML-LSCs, although they are also expressed in normal cells, including activated T cells (line 382-384).
- Provide a simple illustration describing the immune status and the associated factors and tumor microenvironment in patients with CML.
According to the reviewer’s comment, I have added a simple figure describing the several immune factors surrounding CML stem cells, which will clarify the content of the review (Figure 2).
- In paragraph 2.3 describing the TKI for Bcr-Abl, please provide a table with relevant information regarding its immunomodulatory potential and effects.
I have included a table showing immunomodulatory effects of TKIs (Table 1).
- I would like to know whether the combination of TKIs and immunotherapy is a good treatment strategy; if a patient harbor mutant Bcr-Abl, is combinatorial immunotherapy a potentially useful approach? This may also be a topic worthy of discussion, which can be discussed in the revised manuscript.
In case of CML patients with mutant BCR-ABL, strong immunotherapy including CAR-T cells would be available. This is discussed in line388-390 as follows.
TKI-non-responders, TKI-intolerant CML patients, or AP/BC CML patients would be eligible for this therapy.

Reviewer 4 Report
"then every 3 weeks for 1 week," line 260 is this correct?
Author Response
I appreciate the time and effort you and each of the reviewers has dedicated to providing insightful feedback on ways to strengthen my review paper. Thus, it is with great pleasure that I resubmit my article for further consideration. I have incorporated changes that reflect the detailed suggestions you have graciously provided. I also hope that my edits and the responses I provide below satisfactorily address all the issues and concerns you and the reviewers have noted.
‘then every 3 weeks for 1 week," line 260’ is this correct?
The vaccination was conducted every 2 weeks ×4 (Weeks 0, 2, 4, and 6), then on Week 9, and monthly for an additional 10 months, so the sentence is correct, although I added some explanation as follows (line 298-299).
every 2 weeks for 4 weeks (Weeks 0, 2, 4, and 6), then every 3 weeks for 1 week (Week 9), and then monthly for 10 months
